# Zinc as adjunct treatment for clinical severe infection in young infants: A randomized double-blind placebo-controlled trial in India and Nepal

Nitya Wadhwa[1*], Antara Sinha[1], Sudha Basnet[2], Sugandha Arya[3],
Raghvendra Singh[4], Mamta Jajoo[5], Ayushi[1], Dharmendra Sharma[1], Ajay Kumar[6],
Harish K. Pemde[7], Varinder Singh[7], Ram H. Chapagain[8,9], Anuradha Govil[10],
Debjani R. Purakayastha[11], Ganesh P. Shah[12], Medha Mittal[5], Laxman P. Shrestha[2],
Harish Chellani[3], Halvor Sommerfelt[13], Tor A. Strand[13,14*], Shinjini Bhatnagar[1], the
Zinc Sepsis Study Group[¶]

1 Maternal and Child Health, Translational Health Science and Technology Institute, Faridabad, Haryana, India, 2 Department of Child Health, Institute of Medicine, Tribhuvan University, Kathmandu, Nepal, 3 Department of Pediatrics, Vardhman Mahavir Medical College and Safdarjung Hospital, New Delhi, India, 4 Department of Pediatrics, Maulana Azad Medical College, New Delhi, India, 5 Department of Pediatrics, Chacha Nehru Bal Chikitsalaya, New Delhi, India, 6 Department of Neonatology, Maulana Azad Medical College, New Delhi, India, 7 Department of Pediatrics, Kalawati Saran Children's Hospital, Delhi, India, 8 Medical Department, Kanti Children's Hospital, Kathmandu, Nepal, 9 .National Academy of Medical Sciences, Kathmandu, Nepal, 10 Department of Pediatrics, Kasturba Hospital, Delhi, India, 11 Indian Council of Medical Research, New Delhi, India, 12 Department of Pediatrics, Patan Academy of Health Sciences, Lalitpur, Nepal, 13 Centre for Intervention Science in Maternal and Child Health and Centre for International Health, University of Bergen, Bergen, Norway, 14 Department of Research, Innlandet Hospital Trust, Lillehammer, Norway

¶ Membership of the Zinc sepsis study group is listed in the S1 Appendix.
* nitya.wadhwa@thsti.res.in (NW); tors@me.com (TAS)

## Abstract

### Background

Annually, an estimated 2.3 million infants die within their first month of life, primarily in sub-Saharan Africa and South Asia. Infections, including sepsis are among the major contributors to these deaths. Effective interventions added to standard antimicrobial therapy can reduce sepsis mortality. A recent meta-analysis suggests that adjunct zinc treatment of young infants with sepsis could reduce case fatality risk. This study evaluated the efficacy of zinc as an adjunct to antibiotics in young infants with suspected sepsis, defined as clinical severe infection (CSI).

### Methods and findings

We conducted a randomized, double-blind, placebo-controlled trial across seven hospitals in India and Nepal from February 28, 2017, to February 22, 2022. Infants aged 3–59 days hospitalized with suspected sepsis, defined as CSI, adapted from the WHO Integrated Management of Childhood Illness (IMCI) criteria, were randomly assigned to receive 10 mg of elemental zinc daily or placebo orally for 14 days, in addition to standard

**Data availability statement:** All data used in this article will be made available upon request in compliance with the national policies. To meet ethical requirements for the use of confidential patient data, requests must be approved by the relevant institutional ethical review committees. Requests for data should be directed to the Translational Health Science and Technology Institute (THSTI) at dm@thsti.res.in, or the Department of Global Health and Primary Care at the University of Bergen at post@igs.uib.no. For data collected in Nepal, requests should be addressed to the Nepal Health Research Council (NHRC) at nhrc@nhrc.gov.np.

**Funding:** This work was supported by the Research Council of Norway's, Global Health and Vaccination Research (GLOBVAC) program (project number 234505 to SB), the Centre for Intervention Science in Maternal and Child Health (CISMAC; RCN project number 223269 to NW, TS and SU) which is financed by the Research Council of Norway's Centres of Excellence scheme (rcn.no) and the University of Bergen (uib.no), and Innlandet Hospital Trust (Sykehuset-innlandet.no), Norway. The funders had no role in study design, data collection and analysis, decision to publish, or preparation of the manuscript.

**Competing interests:** The authors have declared that no competing interests exist.

**Abbreviations:** CFR, case fatality risk; CI, confidence interval; CRP, C-reactive protein; CSI, clinical severe infection; HR, hazard ratio; IECs, Institutional Ethics Committees; IQR, interquartile range; RCTs, randomized controlled trials; RD, risk difference; RR, relative risk; SD, standard deviation; THSTI, Translational Health Science and Technology Institute; WHO, World Health Organization.

of care. The primary outcomes were death during hospitalization and death within 12 weeks after enrollment. Among 3,153 enrolled infants (1,203 [38%] females), the median age at enrollment was 25 days (interquartile range 13–41 days), and the mean weight was 2.9 kg (standard deviation 0.8). During the hospital stay, 64 (4.1%) of 1,576 infants died in the zinc arm compared to 77 (4.9%) of 1,577 in the placebo arm (relative risk [RR] 0.83 (95% CI [0.60, 1.15]; $p = 0.267$)). Among those who completed 12 weeks of follow-up, 140 of 1,554 infants (9.0%) died in the zinc arm, and 133 of 1,550 (8.6%) in the placebo arm (RR 1.05 (95% CI [0.84, 1.32]; $p = 0.674$)). Adverse events were similar across trial arms, except for a slight increase in vomiting in the zinc arm; no events were attributed to the intervention. The main limitation of the study is that it was underpowered due to lower-than-anticipated event rates and a shortfall in the achieved sample size.

## Conclusions

In this setting, we found little evidence for an effect of adjunct zinc therapy on young infants with CSI on the risk of dying during hospitalization or for the subsequent 3 months. Our findings contrast previous studies that used more specific case definitions. This underscores the need for further RCTs to evaluate the effect of zinc in young infant sepsis before it can be recommended in treatment guidelines.

## Trial registration

Clinical Trials Registry-India (CTRI/2017/02/007966) on February 27, 2017, and Universal Trial Number is U1111-1187-6479.

---

## Author summary

### Why was this study done?

- Almost half of deaths among children less than 5 years of age occur during the neonatal period, i.e., the first 28 days of life. Severe infections contribute to nearly a quarter of these deaths and are among the leading causes of infant hospitalization.

- Limited access to effective antibiotics for serious infections in newborns underscores the importance of evaluating inexpensive and potentially beneficial adjunct treatments such as zinc in low-resource settings.

- The evidence for oral zinc as adjunct treatment for severe infections in young infants is insufficient.

### What did the researchers do and find?

- This large, multisite randomized controlled trial (RCT) was implemented to evaluate oral zinc as an adjunct to standard antibiotic therapy in reducing the risk of death among young infants with clinical severe infection (CSI).

- Infants aged 3–59 days (*n* = 3,153) hospitalized with signs of CSI were randomly assigned to receive 10 mg of oral elemental zinc or placebo daily for 14 days, in addition to the standard of care.

- Among the enrolled infants, 64 infants (4.1%) who received zinc and 77 infants (4.9%) who received placebo died during hospitalization.

- In young infants with at least one sign of CSI, zinc administration did not show a statistically significant benefit on survival.

## What do these findings mean?

- The absence of a beneficial effect of adjunct zinc on survival in young infants with CSI, as observed in this study, does not support its routine use in this population.

- While some previous studies suggest zinc may reduce mortality in young infant sepsis, further large-scale RCTs in diverse settings are needed before integrating it into standard treatment guidelines.

- One limitation of this study is that fewer infants were enrolled than planned, and fewer deaths occurred than expected, which reduced the ability to detect small differences between the groups.

## Introduction

Neonatal mortality remains a major global health concern, with the highest burden in sub-Saharan Africa and South Asia [1]. Of the estimated 2.3 million neonatal deaths each year [1], a significant proportion is attributed to severe infections, including sepsis [1–8]. Antimicrobial resistance to inexpensive first-line antibiotics is common [9,10], and effective antibiotics are often inaccessible for vulnerable children in low-resource settings. It is, accordingly, important to identify inexpensive, effective, and accessible interventions that can be used in addition to antimicrobial and supportive therapy to improve outcomes in severe infections.

Zinc given orally reduces the duration and severity of acute diarrhea in children aged 6–59 months [11–13] and is therefore recommended for its treatment in this age group [14]. Zinc is essential for the normal functioning of the immune system and might also have a role in other infections, including severe bacterial illness [15,16]. In a recent meta-analysis of randomized controlled trials (RCTs) involving infants under 4 months of age treated for young infant sepsis, zinc substantially reduced the risk of treatment failure [17]. Furthermore, in studies providing doses of at least 10 mg elemental zinc per day, this adjunct therapy reduced case fatality risk (CFR) [17–20]. In the largest RCT included in this meta-analysis, 10 mg of daily zinc treatment demonstrated a 40% (95% confidence interval (CI): 10%, 60%) efficacy against treatment failure [18]. The results from this study also suggested a similar effect on the risk of dying [18]. While promising, the evidence is insufficient to recommend zinc as adjunct therapy to young infants with sepsis. We conducted a multicenter RCT to estimate the efficacy of adjunct oral zinc in reducing case fatality among young infants aged 3–59 days with suspected sepsis, defined as clinical severe infection (CSI) adapted from WHO Integrated Management of Childhood Illness (IMCI) guidelines [21–23].

## Methods

### Study design and participants

This was an individually randomized, double-blind placebo-controlled parallel arm superiority trial that was undertaken in five hospitals in India and two in Nepal between February 28, 2017, and February 22, 2022. Ethical approvals were obtained from the Translational Health Science and Technology Institute (THSTI) Institutional Ethics Committee (Ref: THS 1.8.1/(48)); the Nepal Health and Research Council (Ref: 598); the Regional Committee for Medical and Health Research

Ethics in Norway (Ref: 2016/407 REK sør-øst D); and the Institutional Ethics Committees (IECs) of all participating hospitals, including Maulana Azad Medical College and Lok Nayak Hospital (Ref: F.1/IEC/MAMC/(54/03/2016/No/137), Vardhman Mahavir Medical College & Safdarjung Hospital (Ref: IEC/VMMC/SJH/Project/May/2016/207), Chacha Nehru Bal Chikitsalaya (Ref: F.1/IEC/MAMC/(53/2/2016/No134), Kasturba Hospital (Ref: 1770/MS/KH/16), Lady Hardinge Medical College and Associated Hospitals (Ref: LHMC/ECHR/2019/14), Kanti Children's Hospital (Ref: 192/72/73), Institute of Medicine, Tribhuvan University (Ref: 251(6-11-E)2/073/074), and Patan Academy of Health Sciences (Ref: out1610281110). The trial was prospectively registered with the Clinical Trials Registry-India (CTRI/2017/02/007966) and the International Clinical Trials Registry Platform (U1111-1187-6479). A detailed description of the study procedures is available in the protocol [21]. Written informed consent was obtained from the parents or guardians of eligible infants in the local language. For illiterate parents or guardians, consent was provided in the presence of an impartial witness and documented using a thumbprint. The trial adheres to the Consolidated Standards of Reporting Trials (CONSORT) guidelines [24] (S1 Checklist) and to the CONSERVE statement for trials (S2 Checklist). Two key protocol amendments were implemented in June 2017, shortly after trial initiation: (i) the stabilization window was extended from 8 to 24 h to allow clinically eligible but initially unstable infants additional time to stabilize, and (ii) the exclusion criterion was revised from weight-for-age z-score < −4.5 to weight <1,500 g to avoid excluding low birthweight infants who were at higher risk for poor outcomes, but for whom there was no evidence of increased risk associated with zinc supplementation. The amendments were approved by all relevant IECs prior to implementation.

All infants aged 3–59 days visiting the hospitals' emergency units and having at least one of five signs of CSI (stopped feeding well, severe chest indrawing, axillary temperature ≥38.0 °C or <35.5 °C and movement only when stimulated) [21–23], were screened for eligibility. This definition is adapted from the World Health Organization (WHO) IMCI guidelines and reflects suspected, not microbiologically confirmed, sepsis. To be eligible, infants were also required to have been well at some point from birth until the current episode of illness. Infants were excluded if they had a surgical or other medical condition that precluded administration of the study intervention via oral or nasogastric route; had received zinc supplementation within the previous 48 hours; had received injectable antibiotics for ≥48 h prior to hospitalization; required exchange transfusion; were underweight (weight-for-age z-score < −4.5 or weight <1,500 g) at presentation; or were already enrolled in another clinical study [21]. Infants not allowed oral/nasogastric administration of the study treatment were observed for up to 24 h and reassessed for eligibility. Eligible infants were enrolled after their parents/caregivers provided written informed consent.

### Randomization and masking

Randomization was performed using computer-generated sequences in permuted blocks of eight, by a statistician not otherwise involved in the trial, [21] and was stratified by hospital and the presence or absence of diarrhea. Each enrolled infant was randomly assigned to receive either zinc or placebo in an allocation ratio of 1:1. The dispersible zinc and placebo tablets were procured from Nutriset, Malaunay, France (www.nutriset.fr) and were similar in taste and appearance. They were contained in identically looking blister packs sequentially labeled with a unique number according to the generated randomization sequence. The enrolled infant was assigned four blister packs, each containing 10 tablets of 5 mg zinc or placebo. The participant, investigator, as well as the study staff were unaware of trial arm allocation. Masking was maintained during data collection and analysis.

### Procedures

Once consent was obtained and the infant was randomized, one tablet of zinc or placebo was dissolved in 2.5 ml of expressed breast milk or sterile water and administered to the infant. Subsequent doses were administered every 12 h. Each dose was re-administered once if vomiting occurred within a 30-min observation period. If vomiting occurred after both attempts, that dose was skipped and the next scheduled dose was administered as planned. The treatment was

given via a nasogastric tube to infants not allowed oral feeds and withheld in instances as defined in the protocol [21]. At hospital discharge, the caregiver was advised to continue giving the tablets dissolved in breast milk or clean water twice daily at home to complete 14 days of treatment.

The enrolled infants received standard antibiotics and supportive therapy [21]. Antibiotic therapy followed harmonized, site-specific protocols. First-line treatment included intravenous ampicillin or amoxicillin-clavulanic acid with an aminoglycoside (amikacin or gentamicin). Infants previously treated with injectable antibiotics received a third-generation cephalosporin plus aminoglycoside. Suspected meningitis was managed with cefotaxime or ceftriaxone and amikacin, and staphylococcal infections with cloxacillin or amoxicillin-clavulanic acid; vancomycin was added for suspected staphylococcal meningitis. Antibiotics were given for 7–10 days, extended up to 3 weeks for meningitis. Supportive care (fluids, temperature control, and oxygen) was provided as needed. Treating physicians managed all medical care [21]. The study staff monitored them for clinical features of CSI every 6 h or more often if indicated, until discharge. The first follow-up after discharge occurred at 15 days, when the infant was brought back to the hospital, to assess their health and perform a "residual pill count" to document compliance. Subsequent visits at 6 and 12 weeks were conducted by telephone to collect information on the infant's health, feeding, any illness requiring hospitalization, or other serious adverse events.

The infants were monitored closely for adverse events during the study. All such events were managed and followed up until resolution or outcome and reported as described in the protocol [21] and as per Indian guidelines for biomedical research [25].

All data were collected using electronic case report forms according to a harmonized protocol across all study sites and managed at a central data management center in THSTI [21].

Blood specimens were collected at enrollment, at 48–72 h, and at discharge. The specimens were analyzed using BAC-TEC (Becton Dickinson, NJ) to detect invasive pathogenic bacteria and evaluated using a septic screen [21,26]. Plasma zinc concentrations at baseline and discharge were measured using inductively coupled plasma mass spectrometry in a subsample, employing an in-house acid digestion protocol for measuring the 66Zn isotope (corrected with 89Y) developed using Kinetic Energy Discrimination mode.

## Outcomes

The primary outcomes were case fatality (death until hospital discharge) and extended case fatality (death from enrollment to the end of the 12-week study period). The secondary outcomes were (i) treatment failure defined by the worst of one or more of the following events during initial hospitalization: (a) death; (b) need for life support (mechanical ventilation and/or vasoactive drugs); (c) change in antibiotics for persistence or worsening CSI signs, (ii) recovery defined as cessation of signs of CSI, (iii) death after discharge, and (iv) severe illness requiring hospitalization after discharge until the end of the 12-week study period.

Additional prespecified secondary outcomes, including immunobiological effects, cost-effectiveness analyses, and measures of financial risk protection, will be reported in future publications.

## Quality assurance and quality control

We employed rigorous standardized definitions and manuals for study procedures across the seven sites. Intra- and inter-observer variability within and across sites was minimized by training, re-training, and standardization exercises. We had an independent monitoring team and an auditor that ensured the trial was conducted as per Good Clinical Practice and the protocol. The trial was overseen by a data and safety monitoring board that adjudicated all trial adverse events [21].

## Statistical analysis

Our calculated sample size of 3,762 infants (1,881 per trial arm) was based on an assumed 10% CFR among infants <2 months with CSI, to identify ≥30% relative risk (RR) reduction of death with 95% confidence and 90% power. This was inflated to 4,140 to account for an anticipated 10% attrition [21]. The analyses were done on an intention-to-treat basis

and according to a predefined protocol and statistical analysis plan [21]. Analyses were conducted on a complete case basis, given the minimal loss to follow-up.

Using R version 4.3.0 and STATA (StataCorp LLC, Tx) version 18.0, we computed RRs and risk differences (RDs) for outcomes such as case fatality, extended case fatality, and treatment failure. We used generalized linear models of the binomial family with log and identity links, respectively, for these analyses. We estimated the time to death from enrollment to the end of 12 weeks and the time to cessation of CSI signs during hospitalization using Cox proportional hazards models. Ties were handled using Efron approximation. The mean change in plasma zinc concentration from baseline to discharge was compared between the two trial arms using the Student's $t$ test.

Prespecified subgroup analyses were performed on the primary outcomes of death during hospitalization and in the 12-week study period. These subgroups were based on (i) presence or absence of diarrhea on admission, (ii) study hospital and country, (iii) age at enrollment (3–6 days, 7–28 days, and 29–59 days), (iv) presence of sepsis, either signaled by a blood culture with potentially pathogenic bacteria, or a positive septic screen, i.e., with the presence of any two or more of the following laboratory parameters: total leucocyte count <$5 \times 10^9$ cells/L; absolute neutrophil count <$1.5 \times 10^9$ cells/L; band cell: neutrophil ratio >0.2; micro ESR >15 mm at first hour and C-reactive protein (CRP) levels >10 mg/L [21,26]. Additional post-hoc analyses to explore the effect of zinc on death during hospitalization were conducted on subgroups based on the presence or absence of a CSI sign, convulsion, or CRP concentrations (>10 and ≤10 mg/L).

## Results

We screened 28,301 3- to 59-day-old infants between February 28, 2017, and November 30, 2021, in seven hospitals in India and Nepal. Four of these hospitals enrolled participants throughout the study period; in three hospitals (two in Nepal and one in India), enrollment was terminated because of few eligible participants and outcomes. Among the 12,203 infants with signs of CSI, we excluded 8,635 and of the remaining 3,568 eligible infants, 3,153 were randomized, with 1,576 assigned to receive zinc and 1,577 to placebo (Fig 1). Only 1.6% of enrolled infants (zinc arm: 1.4% [22 of 1,576 infants]; placebo arm: 1.7% [27 of 1,577 infants]) were lost to follow-up (S1 Table) and could not be assessed for both primary outcomes.

The participants in the two trial arms had comparable baseline characteristics (Table 1). The enrolled cohort included 1,203 (38%) females. The median age at enrollment was 25 days (interquartile range [IQR] 13–41 days), the mean weight 2.9 kg (standard deviation [SD] 0.8). The median duration of symptoms before admission into the study was 48 h (IQR 24–72 h). Approximately 40% (1,047 of 2,529) of the babies had CRP >10 mg/L, 24% (726 of 3,089 infants) had a positive septic screen, and we found potentially pathogenic bacteria in 7.6% (208 of 2,742 infants) of the blood cultures. The 14-days' course of study treatment was completed by 99% of participants in both trial arms (S2 Table).

During the hospital stay, 4.1% infants (64 of 1,576) in the zinc arm and 4.9% (77 of 1,577) in the placebo arm died, yielding a CFR ratio of 0.83 (95% CI [0.60, 1.15]; $p = 0.267$); (Table 2). The absolute RD was 0.8% (95% CI [−0.6%, 2.3%]; $p = 0.266$).

During the period from enrollment to the end of the 12-week (84 days) study period, 140 of 1,554 infants (9.0%) died in the zinc arm, and 133 of 1,550 infants (8.6%) died in the placebo arm (RR 1.05 (95% CI [0.84, 1.32]; $p = 0.674$)). Among infants who died, the median time to death until end of the 12-week study period was similar in the two trial arms (18 days [IQR 6–37 days] in zinc arm; 12 days [IQR 4–32 days] in placebo arm; hazard ratio [HR] 1.05 (95% CI [0.83, 1.33]; $p = 0.681$) (Table 2 and Fig 2A).

There were 466 (15%) treatment failures, equally divided between the two trial arms (RR 1.01 (95% CI [0.85, 1.19]; $p = 0.914$)) (Table 2). Of these, 30% (141 of 466 infants) were due to deaths during hospitalization, 26% (121 of 466 infants) due to the need for life support, and 44% (204 of 466 infants) had their antibiotics changed. The risk of severe illness requiring hospitalization, the median duration of hospitalization, and the time until recovery from CSI were similar

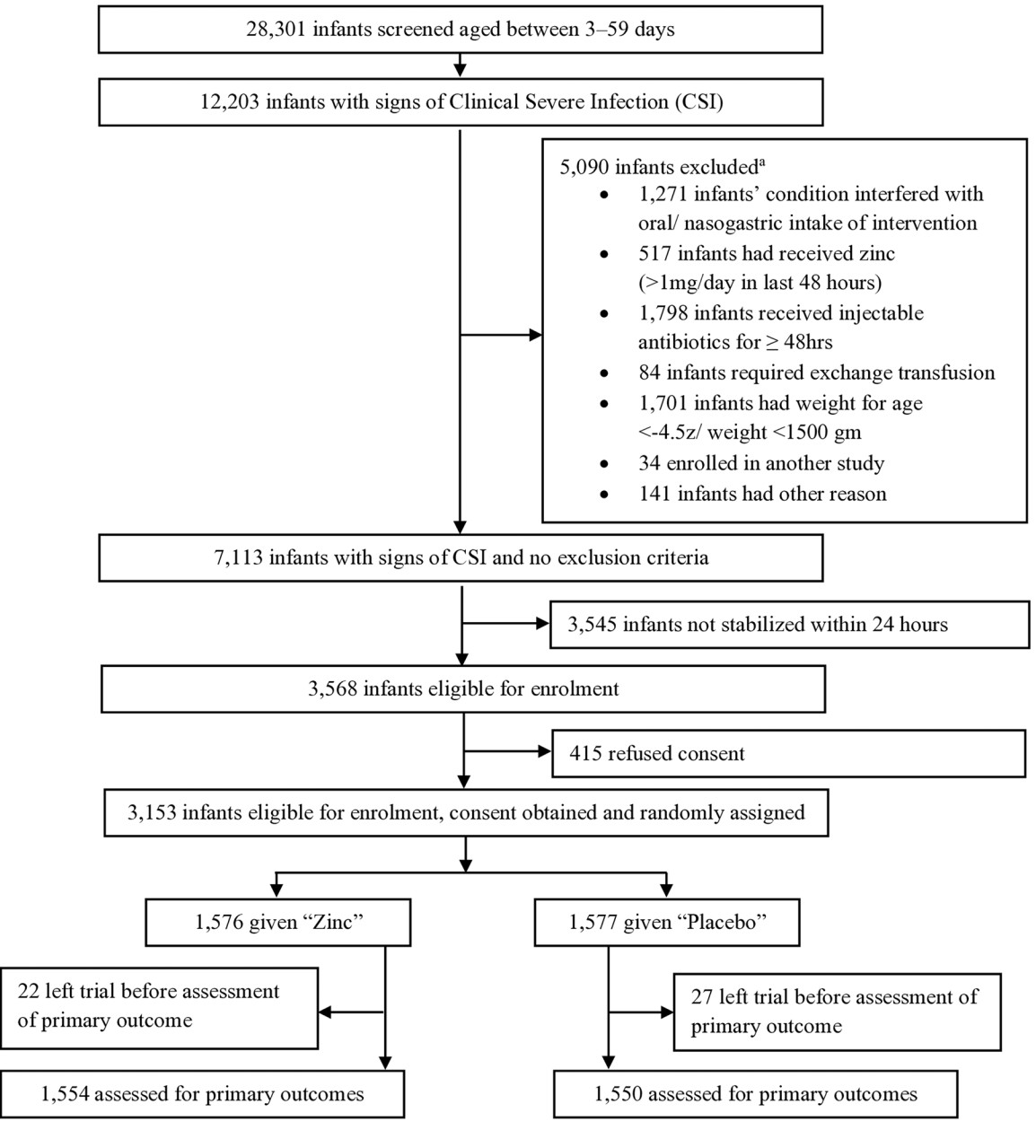

**Fig 1. Study flowchart.**

in the two arms (Table 2 and Fig 2B). The change in plasma zinc concentrations from enrollment to discharge is shown in S3 Table. The mean change in plasma zinc concentration from enrollment to discharge was higher in infants given zinc (2.2 μmol/L [SD 6.2]) compared to those who received placebo (0.3 μmol/L [SD 5.1]), with a mean difference of 1.9 μmol/L (95% CI [−1.3, 5.0]; $p = 0.244$) between the two arms.

**Table 1. Baseline characteristics of study participants.**

| | Zinc (N = 1,576) | Placebo (N = 1,577) |
|---|---|---|
| Age, d[a] | 27 (16) | 27 (16) |
| 3–6[b] | 171 (11%) | 178 (11%) |
| 7–28[b] | 717 (45%) | 693 (44%) |
| 29–59[b] | 688 (44%) | 706 (45%) |
| Female[b] | 614 (39%) | 589 (37%) |
| Weight at enrollment, grams[a] | 2,931 (803) | 2,945 (796) |
| Weight for age z-score[a] | −2.7 (1.6) | −2.7 (1.6) |
| Weight for age <−3z[b] | 627 (39.8%) | 601 (38.1%) |
| Years mother educated[c] | n = 1,574 8 (5–12) | n = 1,575 8 (5–12) |
| Unemployed mother[b] | n = 1,571 1,523 (97%) | n = 1,571 1,518 (97%) |
| **Signs of clinical severe infection** | | |
| Axillary temperature ≥38.0 °C[b] | 486 (31%) | 540 (34%) |
| Axillary temperature <35.5 °C[b] | 10 (0.6%) | 5 (0.3%) |
| Severe chest in-drawing[b] | 584 (37%) | 568 (36%) |
| Stopped feeding well[b] | 1,021 (65%) | 1,003 (63%) |
| Movement only when simulated[b] | 911 (58%) | 887 (56%) |
| **Additional signs** | | |
| Fast breathing[b,d] | 473 (30%) | 429 (27%) |
| Convulsion[b] | 91 (5.8%) | 100 (6.3%) |
| Diarrhea at enrollment[b] | 441 (28%) | 440 (28%) |
| **Laboratory parameters** | | |
| Total Leukocyte Count (× $10^9$ cells/L)[c] | n = 1,419 12.31 (9.3–16.5) | n = 1,426 12.18 (9.1–16.1) |
| Total Leukocyte Count <5 × $10^9$ cells/L[b] | 35 (2.5%) | 55 (3.9%) |
| Absolute Neutrophil Count (× $10^9$ cells/L)[c] | n = 1,373 4.6 (3.1–7.2) | n = 1,357 4.6 (2.9–7.1) |
| Absolute Neutrophil Count <1.5 × $10^9$ cells/L[b] | 56 (4.1%) | 73 (5.4%) |
| Band cells to Neutrophil (IT Ratio)[c] | n = 1,511 0.1 (0.1–0.2) | n = 1,505 0.1 (0.1–0.2) |
| Band cells to Neutrophil >0.2[b] IT Ratio | 331 (22%) | 323 (22%) |
| micro-Erythrocyte Sedimentation Rate (in mm)[c] | n = 1,514 3.0 (2.0–11.0) | n = 1,520 3.0 (2.0–12.0) |
| micro-Erythrocyte Sedimentation Rate >15 mm at first hour[b], mm | 267 (17.6%) | 290 (19.1%) |
| C-reactive protein (mg/L)[c] | n = 1,270 5.9 (1.0–28.7) | n = 1,259 6.3 (1.2–29.2) |
| C-reactive protein >10, mg/L[b] | 523 (41%) | 524 (42%) |
| Septic screen positive[b,e] | n = 1,539 353 (22.9%) | n = 1,550 373 (24.1%) |
| Blood culture: Positive isolates[b] | n = 1,373 111 (8.1%) | n = 1,369 97 (7.1%) |
| Plasma zinc concentration (µmol/L)[c,f] | n = 36 9.1 (8.0–11.0) | n = 36 9.7 (7.7–10.9) |

[a]Mean (SD).

[b]n (%).

[c]Median (IQR).

[d]Respiratory Rate ≥60 breaths/min.

[e]Presence of any two or more of the following laboratory parameters: total leucocyte count <5 × $10^9$ cells/L; absolute neutrophil count <1.5 × $10^9$ cells/L; band cell: neutrophil ratio >0.2; micro erythrocyte sedimentation rate >15 mm at 1 h; C-reactive protein levels >10 mg/L.

[f]Plasma zinc concentration measured at enrollment in a subset of infants from Kalawati Saran Children's hospital.

**Table 2. Effect of adjunct oral zinc on primary and secondary outcomes.**

| Outcome | Zinc[a] (N = 1,576) | Placebo[a] (N = 1,577) | Relative Risk or Hazard Ratio (95% CI)[b] | Risk Difference (95% CI) |
|---|---|---|---|---|
| **Primary** | | | | |
| Death during hospitalization | 64/1,571 (4.1%) | 77/1,573 (4.9%) | 0.83 (0.60, 1.15) | 0.82 (−0.63, 2.27) |
| Death in the 12-week study period | 140/1,554 (9.0%) | 133/1,550 (8.6%) | 1.05 (0.84, 1.32) | −0.42 (−2.42, 1.56) |
| Time to death until end of 12-week study period, d[c] | 18 (6–37) | 12 (4–32) | 1.05 (0.83, 1.33)[d] | |
| **Secondary** | | | | |
| Treatment failure | 234 (15%) | 232 (15%) | 1.01 (0.85, 1.19) | −0.14 (−2.62, 2.34) |
| *Treatment failure (by worst outcome)* | | | | |
| Life support | 67 (4.3%) | 54 (3.4%) | 1.24 (0.87, 1.77) | −0.83 (−2.18, 0.52) |
| Change of antibiotics | 103 (6.5%) | 101 (6.4%) | 1.02 (0.78, 1.33) | −0.13 (−1.85, 1.59) |
| *for worsening of initial clinical signs* | 58 (3.7%) | 59 (3.7%) | 0.98 (0.69, 1.40) | 0.06 (−1.27, 1.39) |
| *for persistence of initial clinical signs* | 45 (2.9%) | 42 (2.7%) | 1.07 (0.71, 1.63) | −0.19 (−1.35, 0.96) |
| Death at any time after discharge until 12 weeks study period | 76/1,490 (5.1%) | 56/1,473 (3.8%) | 1.34 (0.96, 1.88) | −1.30 (−2.78, 0.18) |
| Severe illness requiring hospitalization after discharge | 124 (7.9%) | 122 (7.7%) | 1.02 (0.80, 1.29) | −0.13 (−2.00, 1.74) |
| Time to recovery from clinical severe infection, d[c] | 1.46 (0.77–3.20) | 1.36 (0.75–3.07) | 0.97 (0.90, 1.04)[d] | |

[a]All values are n (%) except where specified.

[b]All the values are relative risks, except where noted.

[c]Median (IQR).

[d]Hazard ratio.

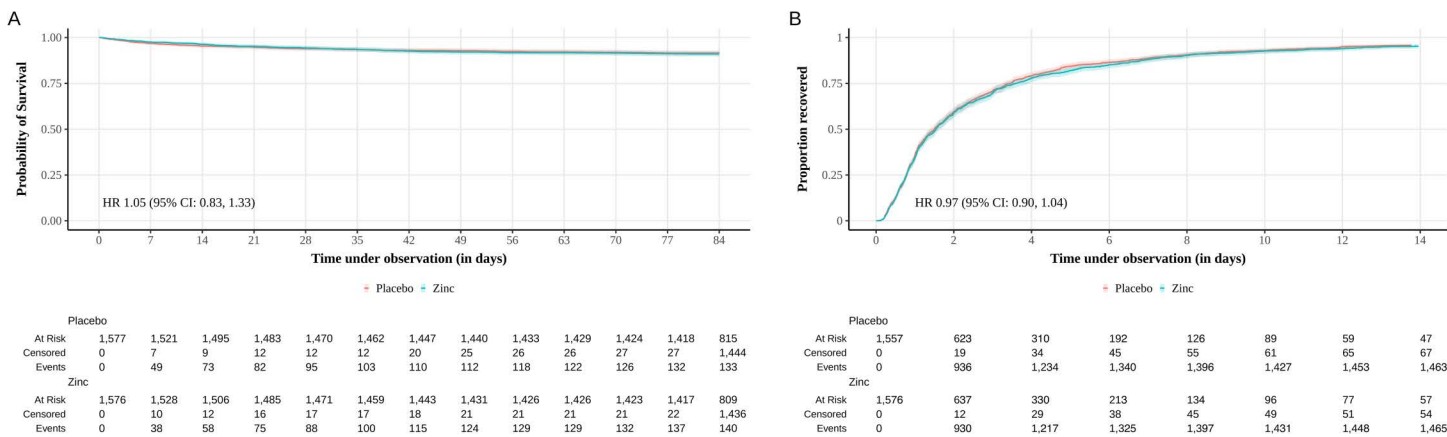

**Fig 2. Kaplan–Meier curves of mortality and recovery in young infants with clinical severe infection.** Time-to-event analyses using a Cox proportional hazards regression model. **(A)** Time-to-death until the end of 12-week study period in the zinc arm (n = 1,576) compared with placebo (n = 1,577). Median (IQR) time to death: zinc, 18 days (6–37 days); placebo, 12 days (4–32 days). Hazard ratio (HR) 1.05 (95% CI [0.83, 1.33]; p = 0.681). **(B)** Time-to-recovery from clinical severe infection in the zinc arm (n = 1,576) compared with placebo (n = 1,577). Median (IQR) time to recovery: zinc, 1.46 days (0.77–3.20 days); placebo, 1.36 days (0.75–3.07 days). HR 0.97 (95% CI [0.90, 1.04]; p = 0.340).

The prespecified and post-hoc subgroup effects of adjunct zinc on the primary outcome of death during hospitalization are depicted in Fig 3A and 3B, respectively. The largest effects were observed in the subgroups of infants who were sepsis positive, among 7-to-28-day-old babies, in infants with CRP > 10 mg/L, and in those who presented with convulsions. All the subgroup-specific effects had wide CIs which included the null effect. The prespecified and post-hoc subgroup

## A

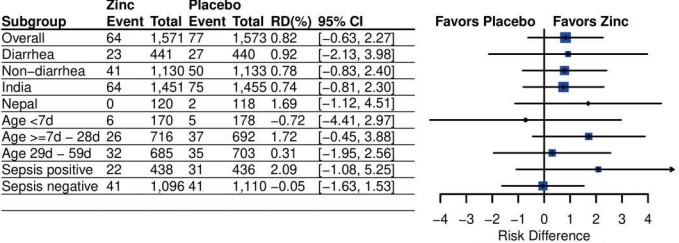

| Subgroup | Zinc Event | Zinc Total | Placebo Event | Placebo Total | RD(%) | 95% CI |
|---|---|---|---|---|---|---|
| Overall | 64 | 1,571 | 77 | 1,573 | 0.82 | [−0.63, 2.27] |
| Diarrhea | 23 | 441 | 27 | 440 | 0.92 | [−2.13, 3.98] |
| Non–diarrhea | 41 | 1,130 | 50 | 1,133 | 0.78 | [−0.83, 2.40] |
| India | 64 | 1,451 | 75 | 1,455 | 0.74 | [−0.81, 2.30] |
| Nepal | 0 | 120 | 2 | 118 | 1.69 | [−1.12, 4.51] |
| Age <7d | 6 | 170 | 5 | 178 | −0.72 | [−4.41, 2.97] |
| Age >=7d – 28d | 26 | 716 | 37 | 692 | 1.72 | [−0.45, 3.88] |
| Age 29d – 59d | 32 | 685 | 35 | 703 | 0.31 | [−1.95, 2.56] |
| Sepsis positive | 22 | 438 | 31 | 436 | 2.09 | [−1.08, 5.25] |
| Sepsis negative | 41 | 1,096 | 41 | 1,110 | −0.05 | [−1.63, 1.53] |

## C

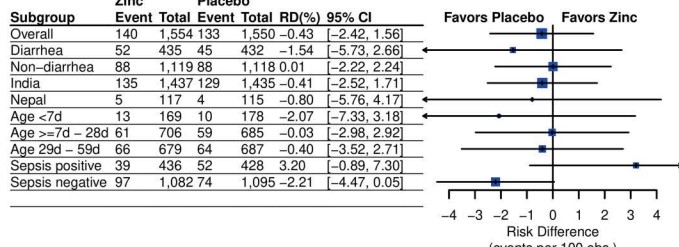

| Subgroup | Zinc Event | Zinc Total | Placebo Event | Placebo Total | RD(%) | 95% CI |
|---|---|---|---|---|---|---|
| Overall | 140 | 1,554 | 133 | 1,550 | −0.43 | [2.42, 1.56] |
| Diarrhea | 52 | 435 | 45 | 432 | −1.54 | [−5.73, 2.66] |
| Non–diarrhea | 88 | 1,119 | 88 | 1,118 | 0.01 | [−2.22, 2.24] |
| India | 135 | 1,437 | 129 | 1,435 | −0.41 | [−2.52, 1.71] |
| Nepal | 5 | 117 | 4 | 115 | −0.80 | [−5.76, 4.17] |
| Age <7d | 13 | 169 | 10 | 178 | −2.07 | [−7.33, 3.18] |
| Age >=7d – 28d | 61 | 706 | 59 | 685 | −0.03 | [−2.98, 2.92] |
| Age 29d – 59d | 66 | 679 | 64 | 687 | −0.40 | [−3.52, 2.71] |
| Sepsis positive | 39 | 436 | 52 | 428 | 3.20 | [−0.89, 7.30] |
| Sepsis negative | 97 | 1,082 | 74 | 1,095 | −2.21 | [−4.47, 0.05] |

## B

| Subgroup | Zinc Event | Zinc Total | Placebo Event | Placebo Total | RD(%) | 95% CI |
|---|---|---|---|---|---|---|
| Overall | 64 | 1,571 | 77 | 1,573 | 0.82 | [0.63, 2.27] |
| Fever | 12 | 486 | 16 | 539 | 0.50 | [1.49, 2.49] |
| No fever | 52 | 1,085 | 61 | 1,034 | 1.11 | [0.81, 3.02] |
| Hypothermia | 2 | 10 | 0 | 5 | −20.00 | [−53.20, 13.20] |
| No Hypothermia | 62 | 1,560 | 77 | 1,568 | 0.94 | [0.51, 2.38] |
| Severe chest–indrawing | 23 | 583 | 29 | 564 | 1.20 | [1.22, 3.61] |
| No severe chest–indrawing | 41 | 988 | 48 | 1,009 | 0.61 | [1.20, 2.42] |
| Stopped feeding well | 37 | 1,015 | 48 | 995 | 1.18 | [0.58, 2.94] |
| Feeding well | 27 | 556 | 29 | 578 | 0.16 | [2.36, 2.68] |
| Movement only on stimulation | 49 | 907 | 55 | 885 | 0.81 | [1.35, 2.98] |
| No movement on stimulation | 15 | 664 | 22 | 688 | 0.94 | [0.80, 2.67] |
| Convulsions | 4 | 90 | 11 | 100 | 6.56 | [0.91, 14.02] |
| No convulsions | 60 | 1,481 | 66 | 1,473 | 0.43 | [1.03, 1.89] |
| CRP >10 | 26 | 522 | 36 | 524 | 1.89 | [0.97, 4.75] |
| CRP <=10 | 24 | 745 | 19 | 732 | −0.63 | [2.34, 1.09] |

## D

| Subgroup | Zinc Event | Zinc Total | Placebo Event | Placebo Total | RD(%) | 95% CI |
|---|---|---|---|---|---|---|
| Overall | 140 | 1,554 | 133 | 1,550 | −0.43 | [2.42, 1.56] |
| Fever | 30 | 482 | 28 | 531 | −0.95 | [3.83, 1.92] |
| No fever | 110 | 1,072 | 105 | 1,019 | 0.04 | [2.56, 2.65] |
| Hypothermia | 2 | 10 | 1 | 5 | 0.00 | [−42.94, 42.94] |
| No Hypothermia | 138 | 1,543 | 132 | 1,545 | −0.40 | [2.39, 1.59] |
| Severe chest–indrawing | 46 | 578 | 54 | 556 | 1.75 | [1.55, 5.06] |
| No severe chest–indrawing | 94 | 976 | 79 | 994 | −1.68 | [4.18, 0.82] |
| Stopped feeding well | 93 | 1,004 | 88 | 982 | −0.30 | [2.83, 2.23] |
| Feeding well | 47 | 550 | 45 | 568 | −0.62 | [3.85, 2.60] |
| Movement only on stimulation | 108 | 898 | 93 | 871 | −1.35 | [4.30, 1.61] |
| No movement on stimulation | 32 | 656 | 40 | 679 | 1.01 | [1.41, 3.43] |
| Convulsions | 9 | 89 | 12 | 100 | 1.89 | [7.05, 10.82] |
| No convulsions | 131 | 1,465 | 121 | 1,450 | −0.60 | [2.64, 1.44] |
| CRP >10 | 50 | 519 | 53 | 516 | 0.64 | [3.01, 4.29] |
| CRP <=10 | 63 | 735 | 45 | 720 | −2.32 | [5.01, 0.37] |

**Fig 3. Subgroup analyses of death in young infants with clinical severe infection.** The figure depicts prespecified and post-hoc subgroup analyses of death during hospitalization and death in the 12-week study period in young infants with clinical severe infection. Risk difference (RD) with 95% CI is derived from generalized linear models of the binomial family with the identity link function. **(A)** Prespecified subgroup analyses for death during hospitalization. Infants were classified as sepsis positive if they had presence of either: (i) positive blood culture with potentially pathogenic bacteria, or (ii) positive septic screen, indicated by any two or more of the following laboratory parameters: total leucocyte count <5 × 10^9 cells/L; absolute neutrophil count <1.5 × 10^9 cells/L; band cell to neutrophil ratio >0.2; micro ESR >15mm in the first hour; and C-reactive protein levels >10 mg/L. **(B)** Post-hoc subgroup analyses for death during hospitalization by baseline characteristics. **(C)** Prespecified subgroup analyses for death in the 12-week study period. Infants were classified as sepsis positive if they had presence of either: (i) positive blood culture with potentially pathogenic bacteria, or (ii) positive septic screen, indicated by any two or more of the following laboratory parameters: total leucocyte count <5 × 10^9 cells/L; absolute neutrophil count <1.5 × 10^9 cells/L; band cell to neutrophil ratio >0.2; micro ESR >15mm in the first hour; and C-reactive protein levels >10 mg/L. **(D)** Post-hoc subgroup analyses for death in the 12-week study period.

analyses for the 12-week mortality outcome presented in Fig 3C and 3D, respectively, indicated no beneficial effects of adjunct zinc. The subgroup analyses for treatment failure are shown in S1 and S2 Figs.

Except for a slight increase in the risk of vomiting, we found no other negative effects of zinc (S4 Table), and no adverse events were considered attributable to the intervention

## Discussion

In this RCT, we enrolled 3,153 young infants treated for CSI in seven hospitals in India and Nepal. We report a small, and statistically not significant, effect of daily adjunct zinc therapy on case fatality. When restricting the analysis to the participants with elevated CRP or a positive septic screen, the observed beneficial effect of zinc against case fatality during hospital stay was larger. We found no evidence that zinc reduced the extended CFR.

We designed this large trial following evidence from an earlier, smaller study indicating that adjunct zinc therapy in infants aged 7–120 days reduced the risk of treatment failure [18]. In this previous study, CRP ≥12 mg/L was one of the inclusion criteria. To increase external validity, the current study used only clinical criteria for defining CSI. We further

limited the age of the study population to infants aged 3–59 days to better align our findings with the young infant age category defined in WHO IMCI guidelines [22,23]. Having case fatality rather than treatment failure as the primary outcome would also increase the relevance for developing guidelines.

Our study has notable strengths. It enrolled many young infants with CSI in several hospitals in Nepal and India, and compliance with the intervention was excellent. Furthermore, very few (0.3%) patients were lost to follow-up for the primary outcome of death during hospitalization. We employed strict enrollment, care, and follow-up protocols to ensure standardized procedures across all sites. Dedicated staff members were rigorously trained at each site to ensure optimal data quality. While the standard of care may vary across hospitals, protocol-driven antibiotic management and site-specific training minimized inter-site differences. This variability, however, reflects real-world conditions and enhances the generalizability of our findings. In addition, adherence to post-discharge zinc or placebo was closely monitored and documented through pill counts and caregiver reports.

Regular monitoring that enabled early detection of danger signs, followed by prompt and appropriate treatment, probably also saved children from dying. Still, we observed 141 deaths during hospitalization and 273 during the entire observation period. The COVID-19 pandemic posed serious challenges to the implementation of our trial; most sites had to temporarily stop recruitment, which also limited the number of eligible children.

Important limitations of this trial include the limited number of events and the shortfall in achieving the projected sample size. While extremely low attrition minimized the risk of selection bias, the reduced sample size and lower-than-expected event rate resulted in diminished statistical power and an increased risk of type II error. It is also possible that several enrolled children were so severely malnourished that they were unable to derive clinical benefit from the adjunct zinc. Furthermore, it is important to acknowledge the heterogeneity among study participants, arising from differences in illness severity, nutritional status, and the standard of care across hospitals, as a potential limitation in interpreting the findings. While we evaluated the influence of these factors on the effect of zinc (Fig 3), it is possible that other relevant effect modifiers were not captured. The considerable overlap in CIs across subgroups limits the ability to draw definitive conclusions regarding effect modification.

Despite their imprecise estimates, it is noteworthy that the effects in the subgroups in which we observed the largest effect resembled those in the studies included in the previously mentioned meta-analysis [17].

In conclusion, in this study, the observed effects of zinc were small and did not reach statistical significance. While our prespecified and post-hoc subgroup analyses may suggest a potential beneficial effect of zinc supplementation in subgroups of infants with laboratory parameters or clinical signs suggestive of more severe illness, these findings are exploratory and require cautious interpretation due to the wide CIs and the absence of any effects on death during the 12-week follow-up period. We suggest that the effect of zinc in young infant sepsis be explored in other RCTs before considering its inclusion in treatment guidelines.

## Supporting information

**S1 Appendix.  List of collaborators in the Zinc Sepsis Study Group.** This appendix provides the names, affiliations, and highest degrees of the members of the Zinc Sepsis Study Group, listed in alphabetical order.
(DOCX)

**S1 Checklist.  Completed CONSORT 2025 checklist for the trial.** This file details how the trial report adheres to the CONSORT 2025 reporting guidelines. This checklist is licensed under the Creative Commons Attribution 4.0 International License (CC BY 4.0; https://creativecommons.org/licenses/by/4.0/).
(DOCX)

**S2 Checklist.  Completed CONSERVE-CONSORT checklist for the trial.** This file documents the extenuating circumstances, important modifications, impacts, and mitigating strategies applied during the trial in response to the COVID-19 pandemic, following the CONSERVE reporting extension.
(DOCX)

**S1 Table. Reasons for losses to follow-up during the 12-week study period.**
(DOCX)

**S2 Table. Compliance to adjunct zinc or placebo.**
(DOCX)

**S3 Table. Plasma zinc in the two treatment arms; change in zinc concentration from enrollment to discharge.**
Plasma zinc concentration measured at enrollment and at discharge in a subset of infants from Kalawati Saran Children's hospital. The data underlying the values shown in this table can be found in S1 Data.
(DOCX)

**S4 Table. Adverse events in young infants with clinical severe infection.**
(DOCX)

**S1 Fig. Prespecified subgroup analysis of treatment failure.**
(DOCX)

**S2 Fig. Post-hoc subgroup analysis of treatment failure.**
(DOCX)

**S1 Data. Underlying dataset for plasma zinc analyses.** This file provides the de-identified individual-level data used to generate S3 Table, including plasma zinc concentrations at enrollment and discharge for infants in the two trial arms. The dataset is restricted to the subsample from Kalawati Saran Children's Hospital.
(CSV)

**S1 Protocol. Study protocol for the Zinc for Young Infant Sepsis Study.** This file provides the full trial protocol, including study rationale, objectives, design, eligibility criteria, interventions, outcomes, statistical analysis plan, and operational procedures.
(PDF)

## Acknowledgments

We express gratitude to all the young infants and their families who have participated in the trial. We are grateful to doctors and nurses of the recruiting hospitals Vardhman Mahavir Medical College and Safdarjung Hospital (VMMC and SJH), Maulana Azad Medical College, and Lok Nayak Hospital (MAMC and LNH), Chacha Nehru Bal Chikitsalaya (CNBC), Kalawati Saran Children's Hospital (KSCH) and Kasturba Hospital (KH) in North India and Kanti Children's Hospital (KCH) and Patan Academy of Health Sciences (PAHS) in Nepal without whom this trial would not have been possible. We thank Drs KC Aggarwal and Sangeeta Yadav for supporting the trial conduct. This work was made possible with support of the Department of Biotechnology, Ministry of Science and Technology, Government of India. We acknowledge the dedication of all the members of the research team: study physicians, study nurses, clinical and laboratory technicians, monitors, the project management, and data management teams. We thank Mr. Mukesh Juyal for secretarial support. We are deeply grateful to Dr Lovejeet Kaur, Dr Shailaja Sopory, and Ms Preeti for the plasma zinc estimation. We thank Dr Siddarth Ramji for his technical inputs and support. We greatly appreciate the monitoring and expert guidance of Dr Jose Martines and the independent oversight provided by the data and safety monitoring board members, including Prof. Vinod K Paul (Chair), Prof. Ravindra Mohan Pandey (statistician), Prof. Anand Prakash Dubey (safety monitor), Prof. Dheeraj Shah, and Prof. Sandeep Agarwala for India and Prof. Neelam Adhikari (Chair), Prof. Kedar Baral, Prof. Sabina Shrestha, Dr Prasanna Samuel Premkumar (statistician) and Mr. Balkrishna Khakurel for Nepal.

## Author contributions

**Conceptualization:** Nitya Wadhwa, Halvor Sommerfelt, Tor A. Strand, Shinjini Bhatnagar.

**Data curation:** Nitya Wadhwa, Antara Sinha, Sudha Basnet, Dharmendra Sharma, Halvor Sommerfelt, Tor A. Strand, Shinjini Bhatnagar.

**Formal analysis:** Nitya Wadhwa, Antara Sinha, Sudha Basnet, Raghvendra Singh, Mamta Jajoo, Ayushi, Dharmendra Sharma, Ajay Kumar, Harish K. Pemde, Varinder Singh, Debjani R. Purakayastha, Medha Mittal, Laxman P. Shrestha, Harish Chellani, Halvor Sommerfelt, Tor A. Strand, Shinjini Bhatnagar.

**Funding acquisition:** Nitya Wadhwa, Sudha Basnet, Halvor Sommerfelt, Tor A. Strand, Shinjini Bhatnagar.

**Investigation:** Nitya Wadhwa, Antara Sinha, Sudha Basnet, Sugandha Arya, Raghvendra Singh, Mamta Jajoo, Ajay Kumar, Harish K. Pemde, Varinder Singh, Ram H. Chapagain, Anuradha Govil, Debjani R. Purakayastha, Ganesh P. Shah, Medha Mittal, Harish Chellani.

**Methodology:** Nitya Wadhwa, Halvor Sommerfelt, Tor A. Strand, Shinjini Bhatnagar.

**Project administration:** Nitya Wadhwa, Sudha Basnet, Raghvendra Singh, Mamta Jajoo, Ajay Kumar, Harish K. Pemde, Ram H. Chapagain, Anuradha Govil, Harish Chellani, Tor A. Strand, Shinjini Bhatnagar.

**Resources:** Nitya Wadhwa, Antara Sinha, Sudha Basnet, Sugandha Arya, Raghvendra Singh, Mamta Jajoo, Dharmendra Sharma, Ajay Kumar, Harish K. Pemde, Varinder Singh, Ram H. Chapagain, Anuradha Govil, Debjani R. Purakayastha, Ganesh P. Shah, Medha Mittal, Harish Chellani.

**Software:** Nitya Wadhwa, Ayushi, Dharmendra Sharma, Tor A. Strand.

**Supervision:** Nitya Wadhwa, Sudha Basnet, Laxman P. Shrestha, Harish Chellani, Halvor Sommerfelt, Tor A. Strand, Shinjini Bhatnagar.

**Validation:** Nitya Wadhwa, Sudha Basnet, Halvor Sommerfelt, Tor A. Strand, Shinjini Bhatnagar.

**Visualization:** Nitya Wadhwa, Antara Sinha, Ayushi, Dharmendra Sharma, Tor A. Strand.

**Writing – original draft:** Nitya Wadhwa, Antara Sinha, Sudha Basnet, Ayushi, Dharmendra Sharma, Laxman P. Shrestha, Halvor Sommerfelt, Tor A. Strand, Shinjini Bhatnagar.

**Writing – review & editing:** Nitya Wadhwa, Antara Sinha, Sudha Basnet, Sugandha Arya, Raghvendra Singh, Mamta Jajoo, Ayushi, Dharmendra Sharma, Ajay Kumar, Harish K. Pemde, Varinder Singh, Ram H. Chapagain, Anuradha Govil, Debjani R. Purakayastha, Ganesh P. Shah, Medha Mittal, Laxman P. Shrestha, Harish Chellani, Halvor Sommerfelt, Tor A. Strand, Shinjini Bhatnagar.

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
