## [Editor Report · Decision Letter 0]

27 Feb 2025

Dear Dr Wadhwa,

Thank you for submitting your manuscript entitled "Zinc as adjunct treatment for clinical severe infection in young infants: A randomized double-blind placebo-controlled trial" for consideration by PLOS Medicine.

Your manuscript has now been evaluated by the PLOS Medicine editorial staff and I am writing to let you know that we would like to send your submission out for external assessment.

However, we first need you to complete your submission by providing the metadata that are required for full assessment. To this end, please login to Editorial Manager where you will find the paper in the 'Submissions Needing Revisions' folder on your homepage. Please click 'Revise Submission' from the Action Links and complete all additional questions in the submission questionnaire.

Please re-submit your manuscript within two working days, i.e. by Mar 03 2025 11:59PM.

Once your full submission is complete and the checks have been passed, your paper will be sent out for external assessment.

Kind regards,

Richard Turner, PhD

Consulting Editor, PLOS Medicine

plosmedicine@plos.org

---

## [Decision Letter · Decision Letter 1]

27 May 2025

Dear Dr Wadhwa,

Many thanks for submitting your manuscript "Zinc as adjunct treatment for clinical severe infection in young infants: A randomized double-blind placebo-controlled trial" (PMEDICINE-D-25-00676R1) to PLOS Medicine. The paper has been reviewed by subject experts and a statistician; their comments are included below and can also be accessed here: [LINK]

After discussing the paper with the editorial team and an academic editor with relevant expertise, I'm pleased to invite you to revise the paper in response to the reviewers' comments. We plan to send the revised paper to some or all of the original reviewers, and we cannot provide any guarantees at this stage regarding publication.

We ask that you submit your revision by Jun 12 2025 11:59PM. However, if this deadline is not feasible, please contact me by email, and we can discuss a suitable alternative.

Don't hesitate to contact me directly with any questions (lgaynor@plos.org).

Best regards,

Louise

Louise Gaynor-Brook, MBBS PhD

Senior Editor

PLOS Medicine

lgaynor@plos.org

Comments from the academic editor:

I have reviewed the paper and agree with most reviewers that it is useful information that should get out. It is unfortunate that they couldn't achieve their planned recruitment. Also with the direction of effects and subgroup analyses, one wishes they had planned a larger and tighter study with solid bacteriology and micronutrient analysis. The bulk of the newborns appear to have late onset sepsis and clinically suspected sepsis/pneumonia and so one wishes that there was greater effort at recruiting early and a tighter group of bacterial sepsis. Some analysis by maternal nutritional status, breastfeeding patterns and RSV / influenza screens would have been super helpful.

Comments from the reviewers:

Reviewer #1: General comments:

I found the manuscript to be clearly written. The methods applied are sound and the results well presented.

The interchangeable use of the terms clinically severe infection (CSI) and sepsis is confusing. I would suggest consistently applying CSI. (e.g. lines 60 and 62, line 113 "sepsis like illness))

Please make it clear the term "young infants" refers throughout the manuscript to infants 3 to 59 months of age.

Specific comments:

Line 106: This sentence seems to imply other treatments could replace antibiotics.

Line 109: Specify efficacy limited to children 6 to 59 months of age.

Vomiting: What proportion had repeated vomiting? How was this managed? Were any excluded?

Study limitations: This was a severely malnourished study population (mean wt-for-age Z-score -2.7). This needs to be highlighted with reference to the inability of these infants to respond to the potential immune response of zinc treatment.

Reviewer #2: This is a challenging study to complete and made more difficult and prolonged by the Covid pandemic. Not accomplishing the planned sample size is a limitation and the reduced power is not of great consequence given how far the primary outcomes are from the hypothesized 30% reduction in mortality.

The methods should state the exclusion criteria. The trial profile gives one reason as condition interfered with oral intake. Then why was NG administration not possible?

It would be useful to present the gestational age and weight for gestational age and sex if these data are available.

Clearly using the CSI definition did not identify many children with likely sepsis s indicated by the laboratory findings and the recovery of 50% by day 2 and 75% by day 4. In this population there is clearly no clinically meaningful benefit of zinc. The suggestion of benefit in infants with confirmed or more likely sepsis is consistent with previous studies, but as the authors indicate would need to be confirmed before zinc can be recommended.

Reviewer #3: Statistical review

This paper reports a randomised trial of zinc vs placebo for infection in young infants. The trial was large and is reported well, giving useful data. I only had some minor comments:

1. Abstract: I would recommend that p-values for the primary outcomes are provided as well as the confidence interval: the study was powered to test that hypothesis.

2. Page 10: "(death from enrollment till discharge from hospital)" - I found this a bit confusing due to the two 'from's, perhaps 'hospital discharge' would avoid that.

3. Page 11, outcomes: there were a couple of secondary outcomes on the trial registration page that were not mentioned here - would these be reported in other work? It would be good to mention this if so.

4. Page 11 - could the assumed control event rate be added for the power calculation as well as the between group difference? I note the 10% inflation in sample size for attrition - could it be stated how missing outcome data was handled in the analysis? Presumably it was complete cases, which seems appropriate given the very low actual loss to follow-up.

5. Page 17 - "The median time to death until end of the 12-week study period was similar in the two trial arms (18 days [IQR 6-37 days] in zinc arm; 12 days [IQR 4-32 days]": I think this should be the median time to death amongst those who died (as fewer than 50% of participants died, there's no median estimable from the trial data).

6. Discussion - given the mention of potential benefit from zinc in more severe subgroups, it would be useful to add if any of the subgroup analyses had significant interaction tests in the results section.

James Wason

---

(Please note: not all will apply to your paper, but please check each item carefully)

* Please upload any figures associated with your paper as individual TIF or EPS files with 300dpi resolution at resubmission; please read our figure guidelines for more information on our requirements: http://journals.plos.org/plosmedicine/s/figures. While revising your submission, please upload your figure files to the PACE digital diagnostic tool, https://pacev2.apexcovantage.com/. PACE helps ensure that figures meet PLOS requirements. To use PACE, you must first register as a user. Then, login and navigate to the UPLOAD tab, where you will find detailed instructions on how to use the tool. If you encounter any issues or have any questions when using PACE, please email us at PLOSMedicine@plos.org.

* Data availability: PLOS Medicine requires that the de-identified data underlying the specific results in a published article be made available, without restrictions on access, in a public repository or as Supporting Information at the time of article publication, provided it is legal and ethical to do so.

The Data Availability Statement (DAS) requires revision. Note that a study author cannot be the contact person for the data.

For each data source used in your study:

* If your trial had to undergo important modifications in response to extenuating circumstances such as the COVID-19 pandemic, please complete the CONSERVE-CONSORT checklist and provide in your Supporting Information; (https://www.equator-network.org/reporting-guidelines/guidelines-for-reporting-trial-protocols-and-completed-trials-modified-due-to-the-covid-19-pandemic-and-other-extenuating-circumstances-the-conserve-2021-statement/). When completing the checklist, please use section and paragraph numbers, rather than page numbers.

FIGURES AND TABLES

SUPPLEMENTARY MATERIAL

REFERENCES

RCTs

* PLOS Medicine requires that all trials be prospectively registered in one of registries recognized by WHO. Please ensure that study registration details are included in the Methods section.

* Please structure the Methods section using the following sub-headings: Study design and participants, Randomization and masking, Procedures, Outcomes, Statistical analysis.

* Amendments to the protocol are outlined in detail in the supplementary file but are not described in the Methods section of the main manuscript text. Please incorporate some of this information in the Methods section.

* Confirmation that all amendments were approved by relevant IRBs is lacking in the Methods section of the main manuscript text.

* Please comment on the statistical power of your trial in the main manuscript text. Planned recruitment appeared to be 2070 per arm, but reached ~1500 per arm.

* The following inclusion criterion is missing in the Methods section of the manuscript text: “Infant should have been well at some point from birth till the current episode of illness.”

* Please outline exclusion criteria in the Methods section.

* Please clarify and explain all discrepancies between the paper and protocol. If the outcomes were not prespecified in the protocol, please define them in the Methods (Outcomes section) as post hoc and explain why they were added. Post-hoc comparisons should be presented as hypothesis generating rather than conclusive.

* Please ensure that all prespecified outcomes (primary, secondary, and exploratory) are listed in the Methods/Outcomes section and indicate whether there are outcomes that are not presented in the current report. More specifically:

- Please provide more detail on all secondary outcomes in the Methods section of the manuscript text.

- Results for cessation of signs of clinical severe infection is not reported in the main text.

- Some time-to-event analyses missing in the main text.

- Secondary outcomes relating to reappearance of CSI, discharge, CEA and Immunobiological readouts are not reported and should be outlined in the methods (even if results not reported).

- Subgroup analyses relating to age at enrolment appear post-hoc (3–6 days, 7–28 days, and 29–59 days). Please clarify.

- Subgroup analyses were also planned for secondary outcomes, but have not been included in this report. Please clarify.

- Some per-protocol analyses were planned but are not included in this report. Please clarify.

* Please specify the dates (Month Day, Year) during which study enrollment and follow up occurred.

* Please include absolute numbers wherever you report percentages; eg, n/N (%)

* Please present the safety data for the study including numbers of specific events and whether or not adverse events are thought to be related to treatment. AEs should be reported in the abstract, per CONSORT and CONSORT-Harms.

* Please complete the CONSORT checklist (https://www.equator-network.org/reporting-guidelines/consort/) and ensure that all components of CONSORT are present in the manuscript, including how randomization was performed, allocation concealment, blinding of intervention, definition of lost to follow-up, power statement. When completing the checklist, please use section and paragraph numbers, rather than page numbers.

* Please report your abstract according to CONSORT for abstracts, following the PLOS Medicine abstract structure (Background, Methods and Findings, Conclusions) https://www.equator-network.org/reporting-guidelines/consort-abstracts/

* If your trial had to undergo important modifications in response to extenuating circumstances, please complete the CONSERVE-CONSORT checklist and provide in your Supporting Information; (https://www.equator-network.org/reporting-guidelines/guidelines-for-reporting-trial-protocols-and-completed-trials-modified-due-to-the-covid-19-pandemic-and-other-extenuating-circumstances-the-conserve-2021-statement/). When completing the checklist, please use section and paragraph numbers, rather than page numbers.

* In keeping with our commitment to Open Science, please include the study protocol document and analysis plan (including any amendments) as Supporting Information to be published with the manuscript if accepted.

* Please note that PLOS Medicine requires prospective, public registration of a data sharing plan (as part of mandatory clinical trials registration) for all clinical trials that began enrollment on or after January 1, 2019, in accordance with ICMJE requirements.

---

## [Decision Letter · Decision Letter 2]

30 Jul 2025

Dear Dr. Wadhwa,

Thank you very much for re-submitting your manuscript "Zinc as adjunct treatment for clinical severe infection in young infants: A randomized double-blind placebo-controlled trial" (PMEDICINE-D-25-00676R2) for review by PLOS Medicine.

I have discussed the paper with my colleagues and the academic editor and it was also seen again by 3 reviewers. I am pleased to say that provided the remaining editorial and production issues are dealt with we are planning to accept the paper for publication in the journal.

[LINK]

We look forward to receiving the revised manuscript by Aug 06 2025 11:59PM.   

Sincerely,

Suzanne De Bruijn, PhD

Associate Editor 

PLOS Medicine

plosmedicine.org

Requests from Editors:

GENERAL:

* Please amend your title to "Zinc as adjunct treatment for suspected clinical severe infection in young infants: A randomized double-blind placebo-controlled trial in India and Nepal".

* Please ensure that all abbreviations are defined at first use throughout the text.

* Please confirm that all numbers presented in the abstract are present and identical to numbers presented in the main manuscript text.

ABSTRACT:

* Please include ‘arm’ after ‘zinc’ in the following sentence: “During 12 weeks from enrolment, 140 of 1,554 infants (9.0%) died in the zinc, and 133 of 1,550 infants (8.6%) in the placebo arm (RR 1.05 (95% CI [0.84, 1.32]; p=0.674)).”

* Please include the fact that the study is underpowered as a limitation.

* line 38: are the mentioned deaths attributable to infection? please clarify this.

* Line 43—needs a period.

* Line 48: state whether this was a confirmed or a suspected infection.

* Line 49: State that zinc or placebo was added to SOC (Standard of care).

* Line 53: replace 'against' with 'compared to' or 'and'

* Line 55: Please clarify why the number of total infants is different between the hospital stay vs 12 weeks. (e.g due to loss of follow-up)

AUTHOR SUMMARY:

*Please move the last bullet point in ‘why was this study done’ to ‘what did the researchers do and find.

*Please remove ‘well-designed’ from your author summary: “This well-designed, large, multisite randomized controlled trial (RCT) was implemented to evaluate oral zinc as an adjunct to standard antibiotic therapy in reducing the risk of death among young infants with clinical severe infection (CSI)”

*Please remove the first bullet point in ‘what did the researchers do and find’ (Our study, undertaken at several institutions in two Asian countries, is the largest RCT on the effect of zinc as an adjunct to standard antimicrobial therapy in reducing death during hospitalization and within 12 weeks after enrolment among young infants with CSI.)

* Please remove the last bullet point (line 100-102) under 'What Do These Findings Mean?' (as this is redundant with the previous bullet point)

* In the author summary, in the final bullet point of 'What Do These Findings Mean?', please include the main limitations of the study in non-technical language.

* Please include the fact that the study is underpowered as a limitation.

* Please ensure that all text is objective.

* Please rephrase line 77, which states that zinc is an effective adjunct treatment (which is at odds with the following bullet point in line 79).

* Line 89: Please mention that Zn or placebo was added to SOC.

INTRODUCTION:

* Please have a look at the first 2 lines; these are currently almost identical to the first lines of the Abstract.

* Line 123: consider stating 'suspected sepsis' rather than 'sepsis'.

* Please provide a definition of clinical severe infection in the introduction, and clarify that this is suspected, not microbiologically confirmed. Clarify whether this is a conventional clinical term and/or whether this is used in India and Nepal.

METHODS:

* line 175: do you mean 'sterile' water with 'clean water'? Please specify.

* Line 202: you state that Plasma zinc concentrations at baseline and discharge were measured, and I see these are reported in S3. However, please also provide us with the raw data for this in a supplemental file.

* Please provide more details about the standard of care at each of the different study sites.

DISCUSSION:

* Please include a discussion on the potential differences in SOC across the study sites. This needs to include potential heterogeneity across hospitals, and should comment on adherence of pill taking once patients and caregivers returned home.

* Line 335: please remove 'precisely estimated' from this sentence.

* Please ensure that discussion of the study is objective.

*Please temper any conclusions related to post hoc analyses.

FUNDING STATEMENT:

* The funding statement should include: specific grant numbers, initials of authors who received each award, URLs to sponsors’ websites. Also, please state whether any sponsors or funders (other than the named authors) played any role in study design, data collection and analysis, the decision to publish, or preparation of the manuscript. If they had no role in the research, include this sentence: “The funders had no role in study design, data collection and analysis, decision to publish, or preparation of the manuscript.”

ETHICS AND CONSENT:

*Thank you including the names of the IRBs who provided ethical approval for this study. Please also include IRB approval numbers.

FIGURES:

* Please provide titles and legends for all figures and tables (including those in Supporting Information files). Please define all acronyms used in each figure or table in its corresponding legend. Please ensure that you include a title for each figure, and include the description of each subpanel in the legend. Specifically this needs to be done for Figure 3.

RCT:

*Thank you for completing the CONSORT 2025 checklist and including it as supporting information. Please cite the checklist in the article (e.g. as ‘S1 Checklist’).

* As your trial had to undergo important modifications in response to extenuating circumstances, please complete the CONSERVE-CONSORT checklist and provide in your Supporting Information.

COMMENTS FROM REVIEWERS:

Reviewer #1: I have read the authors' responses to the 3 reviewer comments and find them acceptable. After reading the full revised text I have no further comments to add.

Reviewer #3: Thank you to the authors for addressing my previous comments well. On their last response regarding interaction testing: I would pesonally recommend that the discussion, if it mentions interpretations from subgroup analysis, should caveat it my stating that was not significant evidence for there being an interaction. Other than this, I am happy with the revised paper.

[LINK]

---

## [Editor Report · Decision Letter 3]

26 Aug 2025

Dear Dr. Wadhwa,

Thank you very much for re-submitting your manuscript "Zinc as adjunct treatment for clinical severe infection in young infants: A randomized double-blind placebo-controlled trial in India and Nepal" (PMEDICINE-D-25-00676R3) for review by PLOS Medicine.

Before we can accept your manuscript, we have a few remaining queries:

1) Discussion: you mention the study has ‘notable strengths’. Please signpost the limitations in the same clear way.

2) Please include URLs for the funding agencies in your financial disclosure

3) Title and legends:

- For Figure 1: please change your title to “study flowchart”

-For Figure 2 and 3: Please provide a title for each figure, and a legend, which includes the description of each subpanel. (Currently, there are separate titles for the subpanels)

4) CONSERVE-checklist: Thanks for providing the CONSERVE-checklist. Can you please also provide a citation to this checklist within the manuscript (similar to the “S1 checklist”)?

5) Thank you for providing us with the raw data for the plasma zinc concentrations at baseline and discharge. Could you please upload these as supplemental data, and state in the legend of figure S3 where these data can be found? We suggest the following statement “The data underlying the graphs shown in the figure can be found in S1 data”.

We look forward to receiving the revised manuscript by Sep 02 2025 11:59PM.   

Sincerely,

Suzanne De Bruijn, PhD

Associate Editor 

PLOS Medicine

---

## [Editor Report · Decision Letter 4]

9 Sep 2025

Dear Dr Wadhwa, 

On behalf of my colleagues and the Academic Editor, Zulfiqar Bhutta, I am pleased to inform you that we have agreed to publish your manuscript "Zinc as adjunct treatment for clinical severe infection in young infants: A randomized double-blind placebo-controlled trial in India and Nepal" (PMEDICINE-D-25-00676R4) in PLOS Medicine.

PRESS

Sincerely, 

Suzanne De Bruijn, PhD 

Associate Editor 

PLOS Medicine